# Power Dynamics in 21st-Century Food Systems

**DOI:** 10.3390/nu11102544

**Published:** 2019-10-22

**Authors:** Boyd Swinburn

**Affiliations:** 1School of Population Health, University of Auckland, 1142 Auckland, New Zealand; boyd.swinburn@auckland.ac.nz; 2Global Obesity Centre (GLOBE), Deakin University, Burwood, VIC 3125, Australia

**Keywords:** food systems, power dynamics, civil society, food industry, government policies, accountability

## Abstract

Food systems are central to our very planetary existence, yet they are not fit for purpose in the 21st century because of the enormous damage they do to the environment and human health. Transforming food systems to optimize human health, ecological health, social equity and economic prosperity will require major changes in power dynamics between players to shift the status quo. The purpose of this paper is to assess these power dynamics and the opportunities for the Great Intergenerational Food Transformation (GIFT)—how this current generation in power can transform food systems within one generation for future generations. The current ‘policy inertia’ preventing food policy action is due to the strong opposition from the commercial food sector, the reluctance of governments to regulate and tax, and the lack of demand for policy action from civil society. The translation of the market power of large food industries into self-serving political power is the dominant barrier to action. The most promising systemic lever for holding the major power players (governments and food industries) to account for the GIFT is increasing the power of civil society (including non-governmental organizations (NGOs), researchers, professional societies and the public) to demand changes in the political economy of food.

## 1. Introduction

We are about one fifth into the 21st century and it is now abundantly clear that the food systems which evolved last century through the Green Revolution, massive global population growth, globalization of trade in goods and services, neoliberal economics, and the increasing concentration of market power in the hands of food company oligopolies are now not fit-for-purpose for the challenges we face in this century [1]. Two outcomes of food systems are inescapable—firstly, they are rapidly, and in some cases, irreversibly destroying our natural environments and secondly, they are by far the biggest cause of disease and premature death. Getting our food systems right is the single most powerful lever we have for achieving ecological health and human health.

It is now apparent that the food systems we have created to feed ourselves are destroying the very planet we depend on. Food systems generate up to 30% of greenhouse gas emissions; take 70% of freshwater use; poison waterways with agricultural run-off, especially from nitrogen and phosphorous overuse; accelerate extinctions and biodiversity loss with deforestation, habitat destruction, wetland drainage and overuse of insecticides and herbicides; and overfish 30% of fish stocks with a further 60% fully fished [2].

In shifting from the Millennium Development Goals to the Sustainable Development Goals (SDGs) [3], it has also become clear that the various forms of malnutrition (including undernutrition, micronutrient deficiencies, overweight/obesity and diet-related non-communicable diseases (NCDs)) need to be combined into a single concept of ‘malnutrition in all its forms.’ This is because many low- and middle-income countries (LMICs) are struggling under the burdens of multiple forms of malnutrition. For example, the prevalence of both obesity in women and underweight in girls is high in the Middle East and North Africa (34% and 18% respectively), the Caribbean (28% and 17% respectively), and Southern Africa (34% and 17% respectively) [4]. These apparently opposite ends of the nutrition spectrum are both created by the same national food systems and the political economies that underpin them. It is time to shift our thinking from the silos of starving children who cannot get enough calories and overweight adults who are getting too many calories to the joined-up thinking that a country’s food system is not delivering a sufficient quality and variety of healthy, affordable foods for all its population.

The Global Burden of Disease group combined the three major nutrition risk factors that they measure (high body mass index (BMI), dietary risks for NCDs, and maternal and child undernutrition) into a single risk factor of malnutrition in all its forms for the Lancet Commission on Obesity report [4]. This combined risk accounts for almost 20% of the global disability life-years (DALYs) lost, more than double the burden created by the next biggest risk factors of tobacco use and high blood pressure. In every region and in every country, malnutrition in all its forms is the biggest cause of DALY loss, although the mix of nutrition problems within that overall risk factor differs from low-income countries (larger proportion from undernutrition) to high-income countries (larger proportion from high BMI and dietary risks for NCDs).

Another shift in thinking is needed from the usual linear view of food-supply chains or food-value chains to considering all food-related activities as part of food systems which are complex, dynamic, and adaptive. Millions of moving parts within the food systems are linked through positive and negative feedback loops to maintain a relatively stable whole. In fact, it is quite wondrous that, in most countries, all these moving parts can deliver relatively stable food environments from which people choose their daily diets in the depths of megacities like Shanghai and Mexico City to the outback of Australia and northern outposts of Canada. Of course, food insecurity is still a dominant problem in many African and Asian countries, often due to chronic political corruption and/or incompetence.

The point is we need to apply systems thinking to the problem. Part of that thinking is to realise that all systems perfectly create the outcomes that they are designed to produce. So, if the outcomes are bad, then we need to analyse the design of the food systems, including the underlying political economy they are built on. However, to re-design the food systems for better outcomes, we need to understand the in-built tendency for the system to push back on attempted changes, thus maintaining the status quo. That is what complex, adaptive systems do.

This paper is based on the 2018 Muriel Bell Lecture which I delivered to the New Zealand Nutrition Society. The purpose is to assess the underlying power dynamics which create our food systems and to consider ways in which we might achieve the major transformations needed to get food systems fit for this century. Much of the thinking for these ideas in this lecture arose during the course of the development of the Lancet Commission on Obesity report on the Global Syndemic of obesity, undernutrition and climate change, which I co-chaired [4], although it is also reflected in other major reports on food systems [1,2]. The part of our thinking that I want to focus on is the crucial role that governance plays. This is where power dynamics are most evident—who gets to decide what rules are set and enforced, how the economic incentives and disincentives are applied, and what norms and expectations are placed on people and organisations.

## 2. What Outcomes Do We Want from Food Systems?

In 2005, a group of eminent nutrition scientists reframed nutrition beyond its classical biological discipline to include the social and environmental dimensions of nutrition [5]. This ‘new nutrition science’ did not include the economic dimension but as we now move our thinking from nutrition to food systems, this must be included. In agriculture, the economics have dominated much of the progress in science, albeit from a very productionist viewpoint—how can we get more tonnes per hectare, more efficient processing, and greater profits? The dominant view that shapes the food systems we have is that food is an economic commodity which is traded for financial gain with its consequences on health and the ecosystem very much secondary issues. In recent years, agricultural research and policies have started to take nutrition more seriously (beyond just producing more calories). Climate and environmental sciences are also seeing food systems as a major lever not only for climate change mitigation but also resilience and adaptation to the likely effects of climate change on food security [1,2].

The Lancet Commission on Obesity brought these converging areas together into a Systems Outcomes Framework (Figure 1). We took the widely-used socio-ecological model which has the individual at the centre surrounded by a series of concentric circles of societal influences from micro to macro [6] and turned it inside out. At the centre, we placed the natural systems, upon which we depend, surrounded by the same set of concentric circles of human systems as the socio-ecological model but with governance and power at the centre. The expected outcomes from food systems are described below.

### 2.1. Human Health and Wellbeing

Improved food systems over the course of the 20th century dramatically reduced food insecurity and undernutrition globally, although progress has not been as fast as expected and there are still more than 800 million people chronically undernourished. Improved nutrition has also been in part responsible for dramatic declines in cardiovascular disease in high-income countries, but unfortunately these are still on the rise in many LMICs. Global trends in increased height [7] and life expectancy [8] can also be partly attributed to improved food systems. However, as noted above, malnutrition in all its forms is now by far the biggest cause of human ill health. In particular, the globally increasing prevalence of obesity and type 2 diabetes has been relentless and not even wealthy countries have been able to reverse these trends [9]. The high and increasing burden of mental health disorders are also partly attributable to dietary factors [10].

### 2.2. Ecological Health and Wellbeing

Unlike the mixed story with human health, food systems have been universally detrimental to ecological health as noted above. Agriculture occupies about 40% of global land and livestock alone contribute to about 12–19% of greenhouse gas emissions and much of the pollution of waterways [11]. Food systems need to dramatically transform to restore the ecological systems they have degraded and become environmentally sustainable.

### 2.3. Social Equity

Improved food systems have pulled millions of people from precarious, food-insecure lives into healthy, productive lives. However, with increasing economic inequities, we are seeing increasing food and nutrition inequities. The obesity and diabetes transition has occurred in parallel with the economic transition. Obesity first appears within the wealthy, urban dwellers, but over time the socio-economic gradient reverses and these diseases become much greater burdens within the lower income and rural populations [12,13]. Food system transformation needs build in equity by design.

### 2.4. Economic Prosperity

Since food systems have been principally designed to feed those who can pay, they are heavily oriented towards economic outcomes—profits for people and companies within the food systems and economic growth, exports and productivity for the countries. However, this prosperity has been unevenly distributed and there is now a considerable concentration of global market power held by a handful of large transnational corporations [14]. Small and medium enterprises, smaller landowners, and family food businesses are being squeezed by the increasingly globalized food systems that work to the advantage of the oligopolies.

## 3. The Global Syndemic and Policy Inertia

The Lancet Commission on Obesity report described how obesity, undernutrition and climate change not only interact negatively with each other but also have common origins in the major macro systems of food, transport, land use and urban design [4]. This synergy of pandemics, called The Global Syndemic, is by far the greatest current and future threat to planetary health—a term incorporating the health of humans and all ecological systems. An intrinsic part of the problem of The Global Syndemic is society’s inability to implement the actions needed to address it. This is called policy inertia and it consists of three major elements, elaborated here in relation to policies to improve food systems.

### 3.1. Opposition from Food Industries

The primary barrier to enacting policies to create healthier, more sustainable food systems is opposition from large food industry companies and associations, primarily the processors of ultra-processed foods opposing healthy food policies, and the meat and dairy producers opposing environment/ greenhouse gas emission policies. Naturally, the businesses which are producing foods and beverages which are unhealthy or which are contributing to ecosystem destruction and driving climate change stand to lose profits if corrective policies are enacted. For example, most agricultural subsidies benefit beef and dairy production, which is the most environmentally destructive agricultural sector, or monoculture crops, like wheat, rice, corn, sugar and seed oils, which are the commodity ingredients for unhealthy ultra-processed foods [15]. These industries are also very concentrated in market power and, as noted by the former Director-General of the World Health Organization (WHO), this economic power is readily translated into political power to maintain those subsidies and favourable policy environments [16]. These companies have very deep pockets, as amply illustrated by the highly funded and often savagely fought battles that Coca Cola, PepsiCo and the trade associations they fund put up whenever the possibility of a tax on sugary beverages is proposed [17].

### 3.2. Reluctance of Politicians to Tax and Regulate

There are occasional examples of politicians putting substantial efforts into taking the lead in implementing the ‘hard’ (and effective) policies recommended by the WHO. Dr Guido Girardi, a Senator from Chile, is one prominent example. He worked tirelessly at the political level for well over a decade against severe opposition from the food industry and many of his fellow politicians to enact a suite of food policies including strong warning labels on healthy foods, restrictions on marketing to children, healthy school food policies and taxes on sugary drinks [18]. This, unfortunately, is the exception. Often politicians seem to believe (against all evidence) in the power of market-based and educational solutions to address obesity, they are spooked into inaction by the real or potential industry opposition (so-called regulatory chill), they want to invest their political capital on other issues, or they are inept or corrupt. Of course, combinations apply, but the net result is that inaction predominates, even as the obesity epidemic unfolds before their eyes and their countries support the blueprints for action that WHO puts to the World Health Assembly year after year. While the power to enact policies for healthier, sustainable food systems rests with the politicians, in reality they are usually very reluctant to use those powers.

### 3.3. Lack of Demand for Policy Action from Civil Society

Opinion polls show that there is strong public support for policies like healthy foods in schools, restrictions on marketing unhealthy foods to children and, in many instances, even taxes on sugary drinks [19]. Other groups from civil society, such as health and consumer non-governmental organizations (NGOs), academics, and professional associations, also strongly support such policies. But this is quiet support. Nobody is marching in the streets on these issues. The problem just does not rise high enough on the urgent to-do list for busy ministers dealing with a myriad of competing issues. Civil society organisations are also generally characterized as small, fragmented, poorly funded and uncoordinated. They might have the passion and commitment but they usually do not have sufficient power to challenge the two powerful sectors in the food battles—governments and the commercial food sector. As I will explore later, civil society may just be the sleeping giant that can find its power to demand the policy action needed to improve food systems, diets and obesity rates.

## 4. Systemic Actions to Address the Global Syndemic

Bringing the pandemics of obesity, undernutrition and climate change together as the Global Syndemic forces the focus down to the common underlying drivers within food systems, transport, land use and urban design and the political economies supporting them. This means that actions at the driver levels can have multiple benefits—double-duty or triple-duty actions which can have benefits for two or three of the pandemics in the Global Syndemic. Examples of triple-duty actions are shown in Table 1 [20]. Achieving these actions will require some major disruptive forces as the dynamics of policy inertia noted above will tend to maintain the status quo.

The Lancet Commission on Obesity report had other specific recommendations at a systemic level for national governments, municipal governments, international agencies, researchers, civil society organisations and funders [4,20]. Of note, the Commission did not have specific recommendations for the private sector including food companies. This was much debated within the Commission. While there are many things that committed companies can do to create healthier and more sustainable food systems, ultimately they are constrained by the laws and regulations they must work within (for example the fiduciary imperative to maximize profits for shareholders) and the nature of the market place which prevents them shifting too far away from their competitors. They cannot, by themselves, create the transformations needed within the food systems; that is the responsibility of governments. Food companies and their trade associations can, however, be very supportive of government actions and not use their economic and political power to undermine attempts by governments to reform food systems. If there were to be one recommendation for the food industry it would have been to refrain from using their economic and political muscle to kill policies that might not be in the company’s financial interests but are in the interests of the health of people and their environments. Will Coca Cola ever refrain from fighting against sugary drinks taxes? Not if we have to rely on their public spiritedness to do so—only if they are forced to. Other companies with a stronger commitment to health and sustainability would be great allies for governments willing to take policy actions.

### The Great Intergenerational Food Transformation (GIFT)

At the same time as the Lancet Commission on Obesity report was published, the EAT Lancet Commission published its report on what constitutes healthy eating patterns within planetary boundaries [2]. Both reports called for fundamental changes to food systems—not just fiddling around the edges but real transformation. A powerful narrative is needed to achieve this transformation because it will be an extremely difficult and highly contested process. Viewed another way, climate change will almost certainly impose major food system transformation on us anyway as major weather events, severe droughts, desertification, floods, climate refugees and civil strife disrupt existing food systems, so it would be smarter to redesign food systems in a condition of calmness not calamity.

With the same urgency that the Inter-governmental Panel on Climate Change is placing on action to reduce greenhouse gas emissions, we need a Great Intergenerational Food Transformation or GIFT. This means that the current generation which is in power, needs to transform food systems within one generation to be healthy, sustainable, equitable and prosperous for future generations. For some countries and regions, narratives around food security, food sovereignty or malnutrition may have greater currency, but whatever the narrative, it needs to be inclusive of the many groups and people who care about food in different ways. People passionate about all aspects of food—organic farming, animal welfare, school meals, composting, pesticide reduction, healthy diets, urban agriculture, food waste and so on—need to be able to see their agenda in the narrative so that they can contribute to the wider movement.

## 5. Power and Accountability Systems

Power is important for accountability purposes. Who holds the power to make a difference and who is holding them to account for their actions and inactions? The three main actors (governments, food companies and civil society) all have their roles to play as do other actors such as media, philanthropic funders, investors, and international organisations. Within this mix, where is the demand for the transformation of food systems likely to come from? The Lancet Commission on Obesity believed governments were central since they hold the main levers of power, but that they were unlikely to act in the absence of visible support from other actors.

### 5.1. Government Actions

Governments at the national, state and municipal levels have the powers to set the rules (laws, regulations, policies) within which the food systems operate. They also have large food procurement budgets for schools, armies, prisons, and government agencies. These procurement powers have been used to increase the demand for organic food (e.g., Denmark [21]), food from family farmers (e.g., Brazil [22]), or non-meat school meals (e.g., New York City [23]).

There are instances where governments have shown great leadership in food policies and Chile is the most prominent recent example. In general, this often rests on the long-term dogged commitment of single politicians (such as Chile’s Senator Guido Girardi) and is currently, unfortunately, a rare occurrence. However, as the impact evaluations of Chile’s policies become available and these country case studies get publicized, they may also become normalized. This is certainly happening in Latin America where several countries are pursuing healthy food policies. Laws on mandatory warning signs on unhealthy foods are already implemented in Peru and Mexico and going through the process in several other countries. On the other hand, the Canadian Government, under its new Prime Minister, Justin Trudeau, proposed a whole suite of food policies for healthier food environments but unfortunately, many of these have become stalled due to powerful industry opposition [24].

Some jurisdictional powers rest with municipal governments (and in large countries, states/provinces) and they also have the opportunity to lead on policy and fiscal actions. For a period over a single four-year term of conservative government in Victoria, a large state-wide, systems-based intervention for healthy food environments was enacted in 12 local government areas in the state [25]. This program was gaining substantial momentum before elections at Federal level brought in a conservative government and Victoria elected a Labor government and, unfortunately, both levels of government pulled the funding on prevention, including for Healthy Together Victoria. The experience of Canada and Victoria are instructional on what committed governments can do to pull together large-scale, coherent policies and programs for improved nutrition, but also, just how fragile these strategic policies are to the power plays of the food industry and party politics.

Municipal governments have traditionally engaged with the food system in relation to ensuring food safety through the implementation of regulations, inspections and certification of food service businesses and through the management of food waste. However, cities are starting to take a wider view on food issues, principally through their work on climate change. Collective international organisations such as C40 [26] and the Milan Urban Food Policy Pact [27] are mobilising cities with an interest in food systems around actions on promoting sustainable diets, regenerative agriculture, food distribution and resiliency, food waste, food governance and monitoring systems.

Amsterdam is a city which has taken a local-government-led, city-wide, multi-strategy approach to reducing childhood obesity. This required strong political commitment at the municipal level, allocated funding for implementation and evaluation, and the engagement of many interested stakeholder groups [28]. The prevalence of people who are overweight and of obesity has reportedly decreased from 21% to 18.5% between 2012 and 2015 [28]. Although the analyses with counterfactual rates for comparable cities in the Netherlands have not been reported in the peer-reviewed literature, these are encouraging results from a municipal government initiative.

### 5.2. Food Company Actions

Food companies not only have powers related to what food they produce, manufacture, market and sell, but they also have considerable political powers in how they influence government policies. Food companies which try to shift towards healthy, sustainable foods can do so under the current business as usual conditions but can only go as far as their customers and shareholders will allow them. They will absolutely not do things like carrying front-of-pack nutrition warning labels, put up their prices to dissuade consumption, or internalize the externalities of the negative impacts on health and the environment. The single most powerful thing food industries can do is to support government policy attempts to create healthier, more sustainable food systems and, especially, refrain from undermining them.

Business models for the 21st century will need to involve much stronger accountability systems on companies, especially large multinational corporations that have gained inordinate power over the last 50 years through increased corporation size, market concentration, corporate wealth, and political influence. While financial accountability is enshrined in law in all countries through mechanisms such as quarterly reporting, financial auditing, and transparency in financially-relevant information, their accountability for human health and wellbeing and the environmental impact is only partially regulated through labour laws and environmental protection laws. It is usually the marketing department that manages the declarations of corporate social responsibility covering the positive deeds of the company but omitting, for example, the damage that their products and processes do to human health, greenhouse gas emissions, and the environment through wasteful packaging. Responsibility (one actor’s declaration of actions) needs to be converted into accountability that involves setting a measurable ‘account’ of actions which one actor (e.g., the government) is able to enforce compliance of another actor, the business, to meet that account. This will require a major change of rules in how businesses operate and account for their actions.

### 5.3. Power of Civil Society—Is It the Sleeping Giant?

There are promising signs that the sleeping giant of civil society may be awakening and flexing its muscles to demand policy action from government and the food industry. There are attempts by academic networks to be more active in monitoring systems for accountability. The INFORMAS network (International Network for Food and Obesity/NCD Research, Monitoring and Action Support) covers more than 40 countries and uses common protocols to measure the actions of governments and food companies and their impacts on food environments such as food composition, labelling, costs, and promotions [29]. These studies create the evidence on what is being done, what should be done and the impacts of implemented policies. This is powerful information for advocacy.

Bloomberg Philanthropies has been supporting civil society organizations and researchers in several countries, mainly in Latin America, to organize in order to demand policy action. The tax on sugary drinks and junk food in Mexico was the first big win for this approach. They fund an NGO like a consumer organization to do the communications and create the voice demanding change; they fund researchers to provide the evidence backing the policies and to evaluate their impacts; and they fund groups to support the legislative process—in other words, the social lobbying. The commission felt that this was a viable model for adoption and adaptation for many more countries and called for other philanthropic funders to create a fund of $1 billion to support this approach in 100 countries.

Momentum is also gathering in many countries demanding action on climate change. The school strikes inspired by Greta Thunberg and the growing protests of Extinction Rebellion are two examples to arise from Europe and start spreading globally. Public protests of this nature plus the growing food sovereignty movements such as Via Campesina and People’s Food Sovereignty, certainly change the power dynamics and may be the disruptive forces needed to achieve transformations of the food system.

## 6. Conclusions

The Global Syndemic is the paramount challenge to planetary health, including human health. A central driver for the syndemic are food systems that are not delivering the outcomes needed for the 21st century. The consensus from expert groups and United Nations agencies is that major transformations of food systems are urgently needed so that they can deliver on human health, ecological health, social equity and economic prosperity. All actors need to work to achieve this but they all have different powers and constraints to make their contributions. Government policy leadership is critical to achieving the rules, economic incentives and disincentives and expectations for food systems’ transformation. To counter the undermining influence of vested commercial interests in creating this policy environment, civil society needs to increase its role in demanding policy action and holding the main actors to account for their actions and inactions. While civil society organisations tend to be highly fragmented and underfunded, the experience from Mexico and other countries is that some targeted funding for advocacy for policy outcomes is potentially an effective mechanism for rebalancing the power dynamics that are currently preventing transformative changes to food systems.

## Figures and Tables

**Figure 1 nutrients-11-02544-f001:**
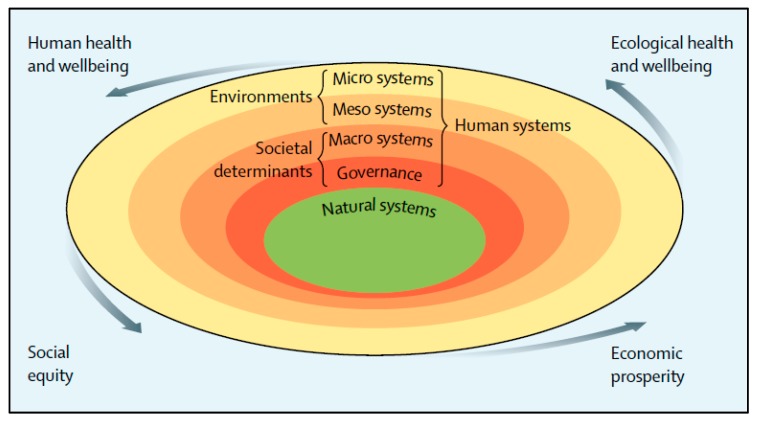
The systems outcomes framework showing the natural systems surrounded by the human systems with the four major consequences coming from these systems (from [4], reproduced with permission from Elsevier 2019).

**Table 1 nutrients-11-02544-t001:** Examples of triple-duty actions for the food systems to benefit action on obesity, undernutrition and climate change [20].

Potential Impacts			
Triple-Duty Action	Obesity/NCDs	Undernutrition	Climate Change
Sustainable dietary guidelines	More healthy, less unhealthy food choices promoted	Improved breast-feeding, healthy food education/access	Decreased demand for unsustainable food choices
Restrict commercial vested interests influence on policy-making	Reduced opposition to obesity/NCD policy implementation	Reduced corruption, more poverty reduction	Reduced opposition to policies to reduce GHG emissions
Reduce red meat consumption	Healthier diets for NCD reduction	More land for efficient, sustainable agriculture	Lower GHG emissions from agriculture
Right to Wellbeing legislation	No marketing of BMS or unhealthy foods	Government requirement to ensure food security for all	Rights of the Child laws include future generations
Framework Convention on Food Systems	Policies enacted for healthy food environments	Policies enacted for poverty reduction and food security	Policies enacted to reduce GHG emissions from food

NCD is non-communicable disease; GHG is greenhouse gas; BMS is breast-milk substitutes.

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
