# Peer review of "Power Dynamics in 21st-Century Food Systems"

_nutrients, 2019, doi:10.3390/nu11102544_

Round 1

Reviewer 1 Report

The article "Power dynamics in 21st century food system" is an attempt to ask and anwer the most crucial questions concerning the current and expected global food system:

Does the current food system meet our needs? What outcomes do we want from food systems? How does the food system affect human health and the natural environment? What action should be undertaken to make the food system effective and environmentally friendly? and finally  Who holds the power to make a difference and who is holding them to account for their actions and inactions? There is no doubt that the described issue is one of the most challenging task for the whole human kind. The article is a very up to date and reasonable analises of the problem.

Author Response

Thank you for those very positive comments - no response required

Reviewer 2 Report

It is an excellent commentary and I don´t think it needs much editing.

However, there is a charge I would like the author to substantiate better. Ln 122-123 author claims "livestock alone 122 contribute to about 19% of greenhouse gas emissions and much of the pollution of waterways". It draws on his own research to justify the claim: Swinburn et al. (2019). Other scholars have offer different conclusions. For example Mitloehner (2018) http://theconversation.com/yes-eating-meat-affects-the-environment-but-cows-are-not-killing-the-climate-94968  Would the author kindly expand his argument to address what other scientists are publishing about the topic?

Also, Ln 151. "Section 3.1. Opposition from food industries" author states "The primary barrier to enacting policies to create healthier, more sustainable food systems is food industry opposition" I would appreciate his view on the role of foodservice industry as well. Would you picture the foodservice activity at the end of the value chain also as a barrier for enacting better food systems or as just an obstacle that can be overcome?

Finally, I take the opportunity to ask you what would be your opinion about the relationship between global obesity pandemic, food environments and food waste (in households, retailers and the foodservice industry)? I do research on this topic but I haven´t found much published about it. I have the impression that these are seemingly related topics that probably are connected but have remained under the radar.

Thanks

Reference

Martin-Rios, C., Demen-Meier, C., Gössling, S., & Cornuz, C. (2018). Food waste management innovations in the foodservice industry. Waste management, 79, 196-206.

Author Response

Thank you for this feedback.

1) L122: The question of the contribution of livestock to GHG production is an important one and one which has several sources to choose from. The reviews points to an article in the Conversation which suggested some flaws in FAO calculations (which they have since corrected), challenges in doing full life cycle analyses for transport as is done for food, and some unhelpful comparisons (like switching between global and US numbers). i found a very useful critique of this Conversation piece rebutting several of the points in the Conversation piece. In my manuscript, it meant to read 'up to 19%' instead of 'to 19%'. I have changed this to the range of '12-19%' which encompasses the revised FAO estimates of 14.5%. We had two IPCC members with expertise in food and greenhouse gases on the Lancet Commission on Obesity, and this range and reference they felt was the best to quote. i have put in the original reference rather than the Commission report which references it.

2) L151: I should have been more specific about which food industries are creating the barrier to action on healthier, more sustainable food systems. In general, the food service industries have been less obstructive to government policies than the large processors of ultra-processed foods (for health-related food policies) and the meat and dairy industries (for environmental agricultural policies). this is now made more specific in the text. Food service industries, such as quick-serve restaurants where unhealthy foods form the majority of their sales, tend to oppose policies, like menu labelling, which affects them directly.

3) Food waste: the reviewer is correct that this topic travels under the radar, including in this manuscript. In line with another reviewer's comment, i have strengthened the part about government action and included food waste within that.

Reviewer 3 Report

This paper is an important and timely commentary on a subject less highlighted within the discourse on sustainable food systems and nutrition. It is nicely written and structured. However, I think that the paper puts great emphasize on defining the problem and the policy inertia, but the reasoning behind the solution provided (civil society) towards the end of the paper is not elaborated enough, and the evidence is weak. While I realize this is a commentary and not a lot of evidence exists, I do think these arguments need further broadening.  

Why is civil society the locomotion for transformation and not the governments? The only answer provided is that government action is a “rare occurrence” and that civil society shows “promising signs”. Maybe reversing government inaction can be a better strategy than awakening civil society? (see the IPES report below on 8 lock-ins). Despite some mention to civil society activities, we are only informed on one example where civil society has been truly successful (Mexico), and this is not convincing enough.

However civil society worldwide has been instrumental in many changes to the food system, and the author should include more examples to strengthen his reasoning, ranging from La Via Campesina to other local grassroots that either provided an alternative or transformed the local food system.

Minor comments:  

I am not sure why the abstract refers to the Muriel Bell lecture. What is it? Why is it detailed in the abstract? It’s just confusing.

Line 35 unnecessary “:”

In table 1, what is the “Framework Convention on Food Systems” .Please also add a reference.

Please cite the IPES report http://www.ipes-food.org/_img/upload/files/UniformityToDiversity_FULL.pdf which touches on many of the themes presented in the commentary.  

Author Response

Thank you for these comments and the request to strengthen the end of the paper on the potential solutions. I have elaborated on the roles of government, business and civil society more to provide the rationale and examples of existing and potential actions to use or change the existing power dynamics to achieve better food system outcomes.

Examples from the IPES report and mentioning other social movements like Via Campesina have been included as suggested.

The reference to the Muriel Bell Lecture has been removed from the abstract but retained in the introduction because I understand there is a tradition of reporting this named lecture in Nutrients as a commentary paper.

Other suggested corrections have been attended to.